# Relationships between Depression, Fear of Missing Out and Social Media Addiction: The Mediating Role of Self-Esteem

**DOI:** 10.3390/healthcare11121667

**Published:** 2023-06-06

**Authors:** Massimiliano Sommantico, Ferdinando Ramaglia, Marina Lacatena

**Affiliations:** 1Dynamic Psychology Laboratory, Department of Humanities, University of Naples Federico II, Via Porta di Massa 1, 80133 Naples, Italy; 2Department of Humanities, University of Naples Federico II, 80133 Naples, Italy; ferdinandoramaglia@gmail.com (F.R.);

**Keywords:** depression, fear of missing out, social media addiction, self-esteem, mediation

## Abstract

The present study examines the relationships between depression, self-esteem, fear of missing out, online fear of missing out, and social media addiction in a sample of 311 Italian young adults (66.2% women and 33.8% men), ages 18–35 yrs. (*M* = 23.5; *SD* = 3.5). The following hypotheses were tested: that depression is positively correlated with fear of missing out, online fear of missing out, and social media addiction, while being significantly negatively correlated with self-esteem; that depression, self-esteem, fear of missing out, and online fear of missing out explain social media addiction scores; that self-esteem mediates the relationship between depression and social media addiction; and that, among Italian participants between the ages of 18 and 35, younger women report higher scores on fear of missing out, online fear of missing out, and social media addiction. Results strongly supported the hypotheses. Taken together, our findings not only contribute to the growing body of research on online addictive behaviors and individuals’ well-being, but also provide support for prevention programs in the field.

## 1. Introduction

Social media has become increasingly popular in recent decades, revolutionizing the way people communicate, work, and entertain themselves. In fact, their use has grown exponentially in recent years. According to the Global Statshot Report 2023 [1], there were an estimated 4.76 billion active social media users worldwide, and in Italy in particular, there were an estimated 43.90 million active social media users. Furthermore, smartphones have also become the main tool for using social media, including, to name but a few, Facebook, Instagram, Twitter, and TikTok. The Global Statshot Report 2023 [1] also indicates that there were an estimated 5.44 billion smartphone users worldwide, and in Italy, there were an estimated 78.19 million smartphone users. The presence and use of smartphones has pushed communication to previously unimaginable limits; in addition to calls, users can send text messages, emails, and voice messages, as well as make video calls, all at extreme distances. 

However, social media has become an important source of information, entertainment, and social connection. The tremendous amount of content available, and the constant presence of an internet connection, can lead users to feel overwhelmed and worried about missing something important. Users can also experience the fear of being separated from their smartphone or not having access to an internet connection. Furthermore, social media users may be constantly exposed to news, events, and activities that they feel compelled to participate in. If they feel that they are not continuously connected, they might miss an important or interesting experience, which can generate anxiety and stress. These feelings are related to constantly experiencing the need, if not the compulsion, to check their social media accounts so as not to miss anything. It is also important to note that social media users can develop an overdependence on the platforms, which is driven by the fear of missing out on something or falling behind their friends and acquaintances, and this can lead to an ongoing obsession with social media as well as a negative impact on mental health, as the previously mentioned constructs highlight.

### 1.1. Fear of Missing Out (FoMO)

Concerning the above considerations, research to date has outlined one construct that accounts for the emergence of real forms of contemporary malaise [2]: the fear of missing out (FoMO) [3,4]. The FoMO is a concept that refers to a state of anxiety experienced by those who use social media; it is caused by the perception that one’s acquaintances are experiencing, or are in possession of, something rewarding that one is not [3,5], which creates a feeling of being excluded or left behind [6]. The construct involves the desire to constantly stay in touch with what others are doing. Research has highlighted four main correlated dimensions: (1) social needs, such as the need to belong to a group [3]; (2) the need to seek approval from others, and accompanying low self-esteem [7]; (3) the emotional problems that emerge in situations where there are difficulties in accessing social media [8]; and (4) social media use which prevents individuals from engaging in daily activities [9]. 

The construct of FoMO thrives in the digital world, especially concerning the use of social networking sites (SNSs), which are seen as privileged channels for maintaining social connections [10] and play a very important role with respect to gratifying social needs [8]. Most notably, these include the need to belong [11] and the need to increase one’s popularity [12]. In this regard, in line with the perspective of self-determination theory (SDT) [13,14], a lack of fulfillment of the aforementioned psychological needs may be related to an increased fear of being missing out on something, which leads to using social networks as a self-regulatory tool to satisfy one’s psychological needs [8]. 

Although the FoMO construct is not yet fully established, and even though its role in the development of maladaptive use of internet communication applications is not yet understood, empirical research has highlighted how it can serve as a mediator between psychopathological symptoms and the consequences of maladaptive use of SNSs on the smartphones [15]. It may also mediate between motivational deficits and social media engagement [16] and between deficits in emotional needs or problems and social media use [3]. FoMO appears to be a predictor of smartphone addiction [17,18] and emotional distress [5], and international research has also highlighted a significant negative correlation between FoMO and self-esteem [19,20,21]. Furthermore, since FoMO appears to be related to online social media use and addiction [22,23,24,25,26,27], Sette and colleagues [28] recently proposed the concept of online fear of missing out (On-FoMO) which, indeed, was strongly positively correlated with psychopathological symptoms and social media addiction [29,30]. 

### 1.2. Social Media Addiction 

Even though the use of modern online technology has repeatedly been associated with positive effects (such as entertainment, social interaction, and the development of cognitive skills, among others), several studies have highlighted relevant concerns about its excessive use [31,32]. In particular, and according to Andreassen and Pallesen [33], the addictive use of online technology is characterized by “being overly concerned about online activities, driven by an uncontrollable motivation to perform the behavior, and devoting so much time and effort to it that it impairs other important life areas” (p. 4054). Although currently not defined as a disorder in the *Diagnostic and Statistical Manual of Mental Disorders* (5th Edition Text Revision; DSM-5-TR) [34] or in the *International Classification of Diseases* (11th Revision; ICD-11) [35], social media addiction could be defined as potentially addictive behavior, with disunity in terminology regarding labels such “excessive use”, “pathological use” or “problematic use”, all concepts referring to social media use disorders. 

However, addictive social media behaviors may also be understood through reference to the six criteria proposed by Griffiths in his model [36]: mood modification (i.e., change in emotional states due to engagement in social media use); salience (i.e., cognitive, emotional, and behavioral preoccupations with the social media use); tolerance (i.e., the ever-increasing use of social media); withdrawal symptoms (i.e., experiencing emotional and physical symptoms due to restricted or discontinued use of social media); conflict (i.e., intrapsychic and interpersonal problems arising from social media use); and relapse (i.e., addicts quickly return to excessive social media use after a period of abstinence). 

Empirical research has highlighted the relationship between addictive technological behaviors, anxiety, depression, and other psychiatric disorders [32,37,38,39]. Research has also demonstrated the role played by specific demographic characteristics. Indeed, younger individuals, as well as individuals not in relationships, tend to more frequently develop social media addiction [31,32,40]. Finally, several studies have found a significant negative correlation between social media addiction and self-esteem [19,41,42,43], thus indicating that addictive social media use is linked to a negative self-concept, and thereby to lower self-esteem. 

### 1.3. Aim and Hypotheses

Considering the above literature review on the topic, the present study aims at analyzing the relationships between depression, self-esteem, FoMO, ON-FoMO, and SMA in Italian young adults. The following hypotheses were tested: that depression is significantly positively correlated with FoMO, ON-FoMO, and SMA, while significantly negatively correlated with self-esteem (H1); that depression, self-esteem, FoMO, and ON-FoMO explain SMA scores (H2); that self-esteem mediates the relationship between depression and SMA (H3); and that women and younger people reported higher scores on FoMO, ON-FoMO, and SMA (H4).

## 2. Method

### 2.1. Procedure and Participants 

Participants were recruited on internet forums and through social media ads from January to March 2023, according to the following criteria: participants must have one or more social media accounts and be between 18 and 35 years old. Participation in the study was anonymous and voluntary, and there was no economic incentive. The rationale, purpose, and procedures of the study were illustrated on the first page of the survey, and participants were asked to sign informed consent to participate in the study. On the second page of the survey, participants were required to complete a basic demographic questionnaire, and the subsequent pages of the survey consisted of a presentation of five different instruments: (1) depression anxiety stress scale-21 (DASS-21) [44]; (2) Rosenberg self-esteem Scale (RSE) [45]; (3) fear of missing out scale (FoMOS) [3]; (4) online fear of missing out (ON-FoMO) [28]; and (5) Bergen social media addiction scale (BSMAS) [39].

The total sample was composed of 311 young adults aged 18–35 years (*M* = 23.5; *SD* = 3.5). Women made up 66.2% of the sample and men 33.8%. Regarding participants’ relationship status, 54.3% were partnered. Regarding participants’ level of education, 64.7% had completed secondary school and 33.8% had completed a university or a post-university degree. Regarding employment, 63% were students, 32.5 were workers, and 4.5 were unemployed. Finally, regarding social media engagement, 15.4% of the participants frequented more than two social media platforms (44.7% frequented two social media platforms), 10.3% spent more than four hours a day on social media platforms (29.3% spent between two and four hours, and 34.4% spent between one and two hours), and 41.5% frequented social media platforms not only during leisure but also during academic and/or work time.

### 2.2. Instruments

#### 2.2.1. Basic Demographic Questionnaire

A basic demographic questionnaire collected information regarding age, gender, educational level, relationship status (single/partnered), occupation, and social media engagement (number of social media platforms frequented, time spent per day on social media platforms, and use of social media platforms during university/work/leisure time).

#### 2.2.2. Depression Anxiety Stress Scale-21 (DASS-21)

The DASS-21 [44,46] is a 21-item self-report instrument assessing depression, anxiety, and stress on three subscales. In this study, the *Depression* subscale was the only one considered (7 items; e.g., “I found it difficult to work up the initiative to do things”). Participants were asked to respond according to a Likert-type scale ranging from 0 (*Did not apply to me at all*) to 3 (*Applied to me very much, or most of the time*). The score, obtained by summing the scores from all seven items and then multiplying by 2, ranges from 0 to 42. The cut-off for a moderate rating of depression is ≥14, while for a severe rating of depression is ≥21. The DASS-21 has demonstrated excellent psychometric properties [44,46], and its internal reliability was excellent in the present study (*α* = 0.94).

#### 2.2.3. Rosenberg Self-Esteem Scale (RSE)

The RSE [45,47] is a 10-item unidimensional self-report instrument assessing self-esteem. Examples of items are: “On the whole, I am satisfied with myself”, “All in all, I am inclined to feel that I am a failure”. Participants were asked to respond according to a Likert-type scale ranging from 1 (*Strongly disagree*) to 4 (*Strongly agree*). The total score was obtained by summing the scores from the ten items ranging from 10 to 40, and higher scores indicate greater self-esteem. The RSE has demonstrated excellent psychometric properties [45,47], and its internal reliability was excellent in the present study (*α* = 0.90).

#### 2.2.4. The Fear of Missing Out Scale (FoMOS)

The FoMOS [3] is a 10-item self-report instrument assessing the fear of missing out. Following the indication for the Italian adaptation of the instrument [48], the FoMOS produces scores on two subscales: (1) *Fear* (FE—4 items; e.g., “I fear my friends have more rewarding experiences than me”) and (2) *Control* (CO—6 items; e.g., “It bothers me when I miss opportunity to meet up with friends”). Participants were asked to respond according to a Likert-type scale ranging from 1 (*Not at all true of me*) to 5 (*Extremely true of me*). The total score was obtained by summing the scores from all the ten items, ranging from 10 to 50, and higher scores indicate greater fear of missing out. The FoMOS has demonstrated good psychometric properties [3,48], and its internal reliability was good in the present study (*α* = 0.84).

#### 2.2.5. Online Fear of Missing Out (On-FoMO)

The On-FoMO [28] is a 20-item self-report instrument assessing online fear of missing out on four subscales: (1) *Need to Belong* (NB—5 items; e.g., “I get annoyed when my friends do not tag me in posts”), (2) *Need for Popularity* (NP—5 items; e.g., “I need people to like or comment on my posts”), (3) *Anxiety* (ANX—5 items; e.g., “I usually feel irritated by staying disconnected from social networks too long”), and (4) *Addiction* (ADD—5 items; e.g., “When I start checking for updates, I find it hard to leave social networks”). Participants were asked to respond according to a Likert-type scale ranging from 1 (*Has nothing to do with me*) to 4 (*Has a lot to do with me*). The total score was obtained by summing the scores from all the twenty items ranging from 20 to 80, and higher scores indicate greater online fear of missing out. The Italian version of the On-FoMO was obtained by performing a translation-back-translation, according to the recommendations of the literature on the cross-cultural adaptation of assessment instruments [49]. The On-FoMO has demonstrated good psychometric properties [28], and its internal reliability was excellent in the present study (*α* = 0.89).

#### 2.2.6. Bergen Social Media Addiction Scale (BSMAS)

The BSMAS [39,50] is a 6-item unidimensional self-report instrument assessing the risk of social media addiction. Examples of items are: “How often during the last year have you felt the urge to use social media more and more?” “How often during the last year have you become restless or troubled if you have been prohibited from using social media?”. Participants were asked to respond according to a Likert-type scale ranging from 1 (*Very rarely*) to 5 (*Very often*). The total score was obtained by summing the scores from all six items, ranging from 6 to 30, and higher scores indicate a greater risk of social media addiction. The BSMAS has demonstrated good psychometric properties [39,50], and its internal reliability was good in the present study (*α* = 0.74).

### 2.3. Data Analyses

Survey data were then entered into SPSS 28.0 [51] database. Internal reliability was verified by means of Cronbach’s *α* that, according to Nunnally and Bernstein [52], was considered satisfactory when values are greater than 0.70. Correlations were calculated by means of Pearson’s coefficient (0.10 < *r* > 0.29 = small association; 0.30 < *r* > 0.49 = medium association; *r* > 0.50 = large association; *p* < 0.05) [53]. Group differences were verified through ANOVA (*p* < 0.05) and *η*^2^ (≥0.01 = small; ≥0.059 = medium; ≥0.138 = large) [53] was used to measure effect sizes. Hierarchical multiple regression analyses were computed using standardized *β* coefficients and *R*^2^ coefficients (*p* < 0.05) to determine the contribution of each predictive variable to the regression models. Finally, mediation analyses were computed through the PROCESS macro tool for SPSS [54], examining both direct and indirect effects with bootstrapping methods to estimate bias-corrected asymmetric confidence intervals (CIs) with 5000 resamples with replacement (CIs not inclusive of zero indicate significant effect) [55].

## 3. Results

### 3.1. Descriptive Statistics

Means, standard deviations, and Cronbach’s *α* for the five instruments are presented in Table 1. The means were: Depression 31.8 (*SD* = 11.3), RSE = 32.3 (*SD* = 6.1), FoMO = 23.7 (*SD* = 7.0), ON-FoMO = 37.5 (*SD* = 10.2), and SMA = 14.4 (*SD* = 4.5).

### 3.2. Correlations and Groups Differences

Results for the total sample (see Table 2) indicated that: participants’ age was significantly negatively correlated with depression, FoMO, ON-FoMO, and SMA; depression was significantly negatively correlated with RSE, while significantly positively correlated with FoMO, ON-FoMO, and SMA; RSE was significantly negatively correlated with depression, FoMO, ON-FoMO, and SMA; FoMO was significantly positively correlated with ON-FoMO and SMA. Similar results emerged when stratified for gender (see Table 3), except for age, which, in the subsample of men, did not correlate with any of the variables included in the study.

These findings strongly supported H1, thus indicating that participants who reported higher levels of depression also reported higher levels of FoMO, ON-FoMO, and SMA, as well as lower levels of self-esteem. These findings also partly supported H4, thus indicating that, among Italian women between the ages of 18–35, younger women reported higher levels of FoMO, ON-FoMO, and SMA.

The ANOVA omnibus test showed statistically significant gender differences. Indeed, women reported higher levels than men for depression (*F*_1, 310_ = 12.07, *p* < 0.01; *η*^2^ = 0.04) (*M_W_* = 33.4; *M_M_* = 28.8), FoMO (*F*_1, 310_ = 4.75, *p* < 0.05; *η*^2^ = 0.02) (*M_W_* = 24.3; *M_M_* = 22.5), ON-FoMO (*F*_1, 310_ = 8.55, *p* < 0.05; *η*^2^ = 0.03) (*M_W_* = 38.7; *M_M_* = 35.1), and SMA (*F*_1, 310_ = 11.10, *p* < 0.01; *η*^2^ = 0.04) (*M_L_* = 15.0; *M_G_* = 13.1), while reported lower levels of RSE (*F*_1, 310_ = 4.49, *p* < 0.05; *η*^2^ = 0.01) (*M_W_* = 31.8; *M_M_* = 33.3).

The ANOVA omnibus test also showed statistically significant differences with respect to occupation. Indeed, student participants reported higher scores than workers on depression (*F*_1, 310_ = 5.50, *p* < 0.01; *η*^2^ = 0.02) (*M_S_* = 33.0; *M_W_* = 29.9), FoMO (*F*_1, 310_ = 9.82, *p* < 0.01; *η*^2^ = 0.03) (*M_S_* = 24.6; *M_W_* = 22.1), and SMA (*F*_1, 310_ = 9.41, *p* < 0.01; *η*^2^ = 0.03) (*M_S_* = 15.0; *M_W_* = 13.4), while they reported lower levels of RSE (*F*_1, 310_ = 4.82, *p* < 0.05; *η*^2^ = 0.02) (*M_S_* = 31.7; *M_W_* = 33.3). 

The ANOVA omnibus and post hoc tests also showed statistically significant differences in social media engagement. Indeed, participants frequenting two or more social media platforms reported higher scores than others on depression (*F*_1, 308_ = 22.38, *p* < 0.01; *η*^2^ = 0.13) (*M_I_* = 28.2; *M_II_* = 32.2; *M_III_* = 40.2), FoMO (*F*_1, 308_ = 16.14, *p* < 0.01; *η*^2^ = 0.09) (*M_I_* = 21.9; *M_II_* = 23.6; *M_III_* = 28.4), ON-FoMO (*F*_1, 308_ = 42.21, *p* < 0.01; *η*^2^ = 0.24) (*M_I_* = 33.3; *M_II_* = 37.4; *M_III_* = 48.2), and SMA (*F*_1, 308_ = 236.25, *p* < 0.01; *η*^2^ = 0.60) (*M_I_* = 11.0; *M_II_* = 15.0; *M_III_* = 21.5), while reported lower levels of RSE (*F*_1, 308_ = 14.25, *p* < 0.01; *η*^2^ = 0.08) (*M_I_* = 34.2; *M_II_* = 31.7; *M_III_* = 29.1). Furthermore, participants frequenting social media platforms for more than four hours per day reported higher scores than others on depression (*F*_1, 307_ = 22.34, *p* < 0.01; *η*^2^ = 0.18) (*M_I_* = 25.8; *M_II_* = 31.1; *M_III_* = 34.4; *M_IV_* = 42.2), FoMO (*F*_1, 307_ = 25.70, *p* < 0.01; *η*^2^ = 0.20) (*M_I_* = 19.3; *M_II_* = 24.1; *M_III_* = 24.8; *M_IV_* = 30.2), ON-FoMO (*F*_1, 307_ = 51.83, *p* < 0.01; *η*^2^ = 0.34) (*M_I_* = 30.1; *M_II_* = 36.7; *M_III_* = 39.7; *M_IV_* = 51.2), and SMA (*F*_1, 307_ = 413.96, *p* < 0.01; *η*^2^ = 0.80) (*M_I_* = 9.1; *M_II_* = 13.4; *M_III_* = 17.5; *M_IV_* = 22.2), while they reported lower levels of RSE (*F*_1, 307_ = 13.34, *p* < 0.01; *η*^2^ = 0.11) (*M_I_* = 34.9; *M_II_* = 32.7; *M_III_* = 31.1; *M_IV_* = 27.8). Finally, participants also frequenting social media platforms during academic and/or work time reported higher scores than participants only frequenting social media platforms during leisure time on depression (*F*_1, 309_ = 38.93, *p* < 0.01; *η*^2^ = 0.11) (*M_I_* = 28.6; *M_II_* = 36.3), FoMO (*F*_1, 309_ = 34.64, *p* < 0.01; *η*^2^ = 0.10) (*M_I_* = 21.8; *M_II_* = 26.3), ON-FoMO (*F*_1, 309_ = 68.57, *p* < 0.01; *η*^2^ = 0.18) (*M_I_* = 33.8; *M_II_* = 42.6), and SMA (*F*_1, 309_ = 322.38, *p* < 0.01; *η*^2^ = 0.51) (*M_I_* = 11.6; *M_II_* = 18.2), while they reported lower levels of RSE (*F*_1, 309_ = 19.82, *p* < 0.01; *η*^2^ = 0.06) (*M_I_* = 33.5; *M_II_* = 32.3).

No statistically significant differences regarding participants’ relationship status and educational level were found. 

### 3.3. Regression Analyses

Based on previous results, hierarchical regression analysis was performed to determine the extent to which each predictor variable contributed to the model explaining SMA above and beyond the others (see Table 4). Considering the high correlations already observed among the predictors, multicollinearity among the independent variables was previously evaluated, and all the Tolerance values were greater than 0.1 (ranging from 0.511 to 0.955). 

For the total sample, age, depression, self-esteem, FoMO, and ON-FoMO, combined, accounted for 64.4% of the variance in SMA and 61.4% of the variance when adjusted for sample size and the number of predictors. Similar results were obtained when conducting hierarchical regression analyses separately for women and men (see Table 4). For the women subsample, age, depression, self-esteem, FoMO, and ON-FoMO, combined, accounted for 61.9% of the variance in SMA and 59.8% of the variance when adjusted for sample size and the number of predictors. For the men subsample, age, depression, self-esteem, FoMO, and ON-FoMO, combined, accounted for 70.6% of the variance in SMA and 68.5% of the variance when adjusted for sample size and the number of predictors.

These findings strongly supported H2, thus indicating that, in the total sample, as well as in subsamples, depression, FoMO, and ON-FoMO positively influenced SMA and explained its scores, as well as that self-esteem negatively influenced SMA and explained its scores.

### 3.4. Mediation Analyses

Based on previous results, we explored the direct and indirect effects of depression on SMA through the variable of self-esteem (see Table 5). 

We found both direct and indirect effects. The coefficient of the direct effect was 0.14 (95% CI [0.05, 0.22]), and the coefficient of the indirect effect was 0.09 (95% CI [0.04, 0.14]). 

These findings strongly supported H3, indicating that depression was associated with SMA and self-esteem. Moreover, the negative coefficient between depression and RSE (−0.32) strongly supports H1, thus indicating that participants reporting higher levels of depression also showed lower levels of self-esteem.

## 4. Discussion

FoMO, ON-FoMO, and addictive social media use have attracted substantial public interest because they are becoming a cornerstone of modern communication, especially among younger people, such as adolescents and young adults. According to international findings [31,32,40,50,56], the results of the present study indicate that, among Italian participants between the ages of 18 and 35, younger women, as well as students, report higher levels of depression, FoMO, ON-FoMO, and SMA. These data are consistent with international literature indicating the higher use of social media communication by “digital natives” who use this channel to develop their social, friendly, and romantic relationships. 

Furthermore, according to the above-cited previous studies [50,56,57,58,59], women reported higher levels of depression, FoMO, ON-FoMO, and SMA. As highlighted by van Deursen and colleagues [40], this gender difference may be interpreted by referring not only to women’s specific social media use, such as maintaining social relationships and gossip, but also to women’s higher anxiety in social interactions and degree of social exposure.

However, regarding age and gender differences, due to mixed results which often result from the poor quality of research on social media addiction behaviors (in terms of sampling, study design, measurement, and cut-off score used), it is difficult to draw statistically significant and sufficiently generalizable conclusions [60,61].

In line with previous studies [3,15,16,22,24,26,27,62], our results indicate a strong association between depression, FoMO, ON-FoMO, and SMA and higher social media engagement, in terms of the number of social media platforms frequented, the number of hours spent daily on social media platforms, and the use of social media platforms during academic and/or work time.

Our results also indicate that FoMO and ON-FoMO are strongly positively correlated with depression and SMA, thus confirming previous research findings [22,23,25,63,64,65]. These data can be interpreted in light of self-determination theory (SDT) [13,14], according to which psychological needs such as autonomy, competence, and relationship can account for motivations for using social media, especially among adolescents and young adults. According to Li and colleagues [27], the frustration of such needs and stronger motivations for social media use could lead to higher levels of FoMO, ON-FoMO, and SMA. 

Moreover, our results are consistent with previous investigations indicating that FoMO, as well as ON-FoMO, and SMA significantly affect individuals’ mental health [15,16,32,37,38,39,58,59,62,66,67]. Indeed, in our sample as well, FoMO, ON-FoMO, and SMA were significantly positively correlated with depression. It is, however, difficult to interpret these data uni-directionally. Indeed, there is evidence from prospective studies on bidirectional relationships, whereby psychopathology can cause SMA, which in turn appears to increase psychopathology. Despite this, FoMO, ON-FoMO, and SMA seem to be revelatory measures associated with negative health outcomes. Indeed, as indicated by previous research findings, users characterized as having higher levels of depression are more likely to use social media to cope with depression and, in turn, they are most likely to engage in excessive social media use [68,69].

Consistent with previous investigations [19,41,42,43,70,71], our results highlighted a significant negative correlation between SMA and self-esteem, which can be explained by referring to the protective role of self-esteem for addictive behaviors, as well as by the higher dependence on others for approval seen in individuals with low self-esteem. In the same vein, we can interpret data regarding the significant negative correlation between self-esteem, FoMO, and ON-FoMO that emerged in our study. Indeed, one of the most frequent uses of social media is for obtaining higher levels of self-esteem, which may be boosted, for example, by the number of “likes” obtained [57,72]. 

Finally, according to all previous research findings [73,74] which highlight the mediating role of self-esteem between psychopathological symptoms and SMA, our results indicated that self-esteem mediates the relationship between depression and SMA. These results indicate that self-esteem is an important protective factor, which thereby mediates the direct influence of depression on SMA, although it does not eliminate its direct effect. This is consistent with previous studies that have found a negative relationship between self-esteem and SMA [19,41,42,43,71,72]; as self-esteem decreases, the risk of SMA increases. 

One specific limitation of the present study is the involvement of a web-based convenience sampling methodology [32]. This form of community-based sampling strategy implies the volunteers’ bias as well as the probable greater social connectedness of participants recruited from social networks. Furthermore, self-selection in online surveys negatively affects representativeness [75]. Moreover, the assessment of all variables of the study by using self-report instruments implies the single-method bias, as well as other limitations related to the self-report methodology. Taken together, these limitations restrict the generalizability of our findings. Furthermore, we can hypothesize that the small or moderate effect sizes reported in the present study stem from the sampling strategy, and future research could implement alternative sampling strategies, such as balancing the sample’s gender composition. Finally, future longitudinal research designs could be outlined, which would also allow for causal inferences to be carried out; that was not possible in this study due to its cross-sectional nature.

## 5. Conclusions

Although the use of the internet and technological communication offers multiple opportunities and benefits, they do not exclude the possibility that some individuals may develop addictive behaviors related to the use of social media. This study focused on depression, FoMO, ON-FoMO, and social media addiction, highlighting the impact of self-esteem through a mediation model, which indicates that the relationship between depression and social media addiction is partly mediated by self-esteem. These findings can inform the implementation of preventive interventions more focused on the importance of working on self-esteem levels in order to limit the effect of depression on social media addiction. Finally, the results indicated that, among Italian participants between the ages of 18 and 35, younger women and students are more at risk of developing social media addictive behaviors, and preventive interventions should be targeted primarily at these populations.

## Figures and Tables

**Table 1 healthcare-11-01667-t001:** Descriptive statistics.

	Women		Men		Total Sample		
	(*N* = 206)		(*N* = 105)		(*N* = 311)		
	*M*	*SD*	*M*	*SD*	*M*	*SD*	*a*
Depression	33.4	11.4	28.8	10.4	31.8	11.3	0.94
RSE	31.8	6.0	33.3	6.2	32.3	6.1	0.90
FoMO	24.3	7.3	22.5	6.2	23.7	7.0	0.84
On-FOMO	38.7	10.3	35.1	9.6	37.5	10.2	0.88
SMA	15.0	4.4	13.2	4.5	14.4	4.5	0.74

**Table 2 healthcare-11-01667-t002:** Correlations between age, depression, RSE, FoMO, On-FoMO, and SMA for the total sample.

	1	2	3	4	5	6
1. Age	-					
2. Depression	−0.14 *	-				
3. RSE	0.08	−0.60 **	-			
4. FoMO	−0.18 **	0.37 **	−0.37 **	-		
5. On-FOMO	−0.13 *	0.34 **	−0.29 **	0.67 **	-	
6. SMA	−0.18 **	0.44 **	−0.34 **	0.49 **	0.67 **	-

** p* < 0.05; ** *p* < 0.01; (*N =* 311).

**Table 3 healthcare-11-01667-t003:** Correlations between age, depression, RSE, FoMO, On-FoMO, and SMA for women and men. (Women *N =* 206; Men *N* = 105).

	1	2	3	4	5	6
1. Age	-	−0.06	0.01	−0.16	0.05	−0.06
2. Depression	−0.18 **	-	−0.55 *	0.42 **	0.31 *	0.41 **
3. RSE	0.10	−0.61 **	-	−0.40 **	−0.27 **	−0.33 **
4. FoMO	−0.18 **	0.32 **	−0.35 **	-	0.56 **	0.46 **
5. On-FOMO	−0.21 **	0.33 **	−0.27 **	0.70 **	-	0.70 **
6. SMA	−0.24 **	0.42 **	−0.33 **	0.50 **	0.64 **	-

Note: Women correlations below the diagonal; men correlations above the diagonal. ** p* < 0.05; ** *p* < 0.01.

**Table 4 healthcare-11-01667-t004:** Hierarchical multiple regression analyses summary for age, gender, depression, RSE, FoMO, and ON-FoMO predicting SMA.

	Total Sample (*N* = 311)					Women (*N* = 206)				Men (*N* = 105)			
Step and Predictor Variable	*β*	*R* ^2^	*adj R* ^2^	*Tol.*	Step and Predictor Variable	*β*	*R* ^2^	*adj R* ^2^	*Tol.*	*β*	*R* ^2^	*adj R* ^2^	*Tol.*
1. Age	−0.183	0.034	0.031 **	0.955	1. Age	−0.183	0.033	0.029 **	0.955	−0.168	0.028	0.022	0.955
2. Gender	0.110	0.046	0.040 *	0.953	2. Depression	0.302	0.122	0.114 ***	0.633	0.408	0.194	0.177 ***	0.694
3. Depression	0.337	0.153	0.145 ***	0.810	3. RSE	−0.233	0.156	0.145 **	0.621	−0.253	0.238	0.214 **	0.692
4. RSE	−0.238	0.190	0.180 ***	0.617	4. FoMO	0.661	0.524	0.511 ***	0.844	0.547	0.466	0.437 ***	0.762
5. FoMO	0.625	0.507	0.496 ***	0.950	5. ON-FoMO	0.379	0.619	0.598 ***	0.511	0.595	0.706	0.685 ***	0.660
6. ON-FoMO	0.453	0.644	0.614 ***	0.645									

* *p <* 0.05, ** *p <* 0.01, *** *p <* 0.001.

**Table 5 healthcare-11-01667-t005:** Mediated outcomes on SMA showing indirect effects of depression through RSE. (*N* = 311).

	Consequent					
	RSE			SMA		
Antecedent	Coefficients	*SE*	*p*	Coefficients	*SE*	*p*
Depression	−0.32	0.02	<0.001	0.14	0.04	<0.001
RSE	-	-	-	−0.27	0.07	<0.001
Constant	42.55	0.84	<0.001	28.05	3.33	<0.001
	*R*^2^ = 0.35			*R*^2^ = 0.17		
	*F*_(1, 309)_ = 169.92, *p* < 0.001			*F*_(2, 308)_ = 31.40, *p* < 0.001		

## Data Availability

The data presented in this study are available on request from the corresponding author.

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
