# Peer review of "Relationships between Depression, Fear of Missing Out and Social Media Addiction: The Mediating Role of Self-Esteem"

_healthcare, 2023, doi:10.3390/healthcare11121667_

Round 1

Reviewer 1 Report

This is an interesting study that tests for a variety of relationships between biological / psychological characteristics and social media activity.  The manuscript has potential, but I have some concerns that I would like the author(s) to address:

1. Nobody is a fan of reviewers who mention items that the author(s) considered out of the scope of the study, but it is problematic that social media engagement by the participants was never described in any way.  How many social media platforms did/do they frequent?  How often?  How long each day?  At work and/or at home?  Professional platforms and/or personal platforms?  An absence of this information does not invalidate the results, but it leaves many questions unanswered and prevents a more nuanced interpretation of the findings.

2. Lines 223 – 233 and Table 2 contain the same information.  It would be more effective to limit the text to the verbal presentation of the findings (sans statistics) and use the table to present the statistics.

3. What is the source for the characterization of correlation coefficients (lines 206-208)?

The manuscript needs attention to the English language presentation.  There are instances that are more than typographical / syntax / diction errors, in that they impact the interpretation.  For example, this is an empirical study, so how does it provide theoretical support for a given position (Abstract)?  Lines 314-315 and other, similar sections also need revision to clarify meaning. 

Author Response

  1. We added the information regarding participants’ social media engagement, as well as related analyses;
  2. We eliminated the statistics, limiting the text to the verbal presentation of the findings of the correlation analyses;
  3. We added the source for the characterization of correlation coefficients;
  4. We revised the English language presentation.

Reviewer 2 Report

In this study, the researchers investigated the connections between depression, self-esteem, Fear of Missing Out (FoMO), Online Fear of Missing Out (oFoMO), and social media addiction among 311 young Italian adults aged 18-35 (66.2% female and 33.8% male), and tested the following hypotheses:

  1. depression having a positive correlation with FoMO, oFoMO, and social media addiction while being negatively correlated with self-esteem; 
  2. depression, self-esteem, FoMO, and oFoMO predicting social media addiction; 
  3. self-esteem mediating the relationship between depression and social media addiction; 
  4. and women and younger individuals scoring higher on FoMO, oFoMO, and social media addiction. 

The tests conducted provided sufficient evidence to support the first and third hypotheses. However, further research is required to validate the conclusions drawn for the second and fourth hypotheses.

Hypothesis #2: depression, self-esteem, FoMO, and oFoMO predicting social media addiction (SMA)

  • Since the goal of this test is to determine the extent each variable is related to SMA for existing data, not to predict individual outcomes for new data, the researchers should change the word 'predicting' to 'explaining' in the hypothesis. 

  • In section 3.3, a stepwise multivariate correlation was conducted to evaluate the contribution of each variable to the model. However, it should be noted that adding a new variable to the model will always increase the accounted variance so an elevated R-square does not mean the new variable is significantly contribute to the model.

  • When conducting a linear regression, multicollinearity among these independent variables should be evaluated, especially considering the high correlations already observed among the predictors in section 3.2.

Hypothesis #4: women and younger individuals scoring higher on FoMO, oFoMO, and social media addiction (SMA)

  • Table 3 presents the correlation between age and FoMo, oFoMo, and SMA. It should be noted that:
    • Age is significant only among women, so younger individual scores higher can only applied to women.
    • To support the finding that younger individuals in women score higher on FoMo, oFoMO, and SMA, the data population should be expanded to include other age groups (i.e., >35). If the author intends to only investigate individuals between the ages of 18-35, then the conclusion should be limited to this age group: among Italian women between the ages of 18-35, younger women score higher on FoMo, oFoMO, and SMA.

good quality

Author Response

  1. According to the reviewer’s suggestions regarding H2, we changed the word ‘predicting’ to ‘explaining’. Moreover, we eliminated the section describing the variables’ contribution to the model, as well as we evaluated the multicollinearity between the independent variables;
  2. Always according to the reviewer’s suggestions regarding H4, only investigating individuals between the ages of 18-35, we limited the conclusions to this age group, by stating that among Italian women between the ages of 18-35, younger women score higher on FoMO, ON-FoMO, and SMA.

Round 2

Reviewer 1 Report

1. In terms of social media usage, what is the unit of time for the provided durations?  Per day?  Per week?

2. The authors still did not address my question about the language in the Abstract.  Given that this was an empirical study, the word "theoretical" needs to be removed from the text.

Minor copy editing.

Author Response

  1. We clarified that the unit of time of social media usage was per day;
  2. We removed the word “theoretical” from the abstract;
  3. A native English-speaking researcher again revised the manuscript and made additional minor changes.

Reviewer 2 Report

See comments for each of the responses author provided:

1. Regarding H2, please provide the Tolerance or VIF for each variable in Table 4, instead of a range (of 0.511 to 0.955) in the manuscript. 0.955 Tolerance almost indicated zero correlation between that independent variable and other independent variables. and even a Tolerance of 0.511 still indicated relatively low multicollinearity. However, some of the variables seems high correlated (i.e. FoMo vs On-FoMo). Additionally, please provide adjusted R^2 rather than incremental R^2 for each additional variable in Table 4. 2. Regarding H4, no further comments.

 Minor editing of English language required

Author Response

  1. We provided Tolerance values for each variable in Table 4, as well as incremental R2;
  2. A native English-speaking researcher again revised the manuscript and made additional minor changes.